

# Meiofauna at a tropical sandy beach in the SW Atlantic: the influence of seasonality on diversity

Gabriel C. Coppo[1], Araiene P. Pereira[1], Sergio A. Netto[2] and
Angelo F. Bernardino[1]

[1] Grupo de Ecologia Bentônica, Universidade Federal do Espírito Santo, Vitória, Espírito Santo,
Brazil
[2] Marítima Estudos Bênticos, Laguna, Santa Catarina, Brazil

Corresponding author
Gabriel C. Coppo,
coppogabriel@gmail.com

## ABSTRACT

**Background:** Sandy beaches are dynamic environments housing a large diversity of organisms and providing important environmental services. Meiofaunal metazoan are small organisms that play a key role in the sediment. Their diversity, distribution and composition are driven by sedimentary and oceanographic parameters. Understanding the diversity patterns of marine meiofauna is critical in a changing world.

**Methods:** In this study, we investigate if there is seasonal difference in meiofaunal assemblage composition and diversity along 1 year and if the marine seascapes dynamics (water masses with particular biogeochemical features, characterized by temperature, salinity, absolute dynamic topography, chromophoric dissolved organic material, chlorophyll-a, and normalized fluorescent line height), rainfall, and sediment parameters (total organic matter, carbonate, carbohydrate, protein, lipids, protein-to-carbohydrate, carbohydrate-to-lipids, and biopolymeric carbon) affect significatively meiofaunal diversity at a tropical sandy beach. We tested two hypotheses here: (i) meiofaunal diversity is higher during warmer months and its composition changes significatively among seasons along a year at a tropical sandy beach, and (ii) meiofaunal diversity metrics are significantly explained by marine seascapes characteristics and sediment parameters. We used metabarcoding (V9 hypervariable region from 18S gene) from sediment samples to assess the meiofaunal assemblage composition and diversity (phylogenetic diversity and Shannon's diversity) over a period of 1 year.

**Results:** Meiofauna was dominated by Crustacea (46% of sequence reads), Annelida (28% of sequence reads) and Nematoda (12% of sequence reads) in periods of the year with high temperatures (>25 °C), high salinity (>31.5 ppt), and calm waters. Our data support our initial hypotheses revealing a higher meiofaunal diversity (phylogenetic and Shannon's Diversity) and different composition during warmer periods of the year. Meiofaunal diversity was driven by a set of multiple variables, including biological variables (biopolymeric carbon) and organic matter quality (protein content, lipid content, and carbohydrate-to-lipid ratio).

## INTRODUCTION

Sandy beaches are the most predominant coastal ecosystems worldwide forming an intricate environment between marine and terrestrial realms, with a large diversity of organisms supporting important biogeochemical processes and providing key ecosystem services (*McLachlan & Brown, 2006*; *Defeo et al., 2009*; *Wu, Meador & Hinrichs, 2018*; *Okamoto et al., 2022*; *Corte et al., 2023*). These environments are dynamic and heavily influenced by global and local oceanographic and physical processes (*e.g.*, granulometric characteristics, wave regime, tides, and currents) (*Di Domenico, Da Cunha Lana & Garraffoni, 2009*), which in turn shape the community structure of these habitats (*Maria, Wandeness & Esteves, 2016*). In addition, sandy beaches are under a range of anthropogenic impacts (including climate change) with signs of declining diversity in numerous areas worldwide (*Bellwood et al., 2004*; *McLachlan & Defeo, 2018*). However, these ecosystems are the least studied coastal environment (*Lercari, 2023*), and understanding how marine diversity varies at local scales and how global and local factors may affect sandy beach biodiversity is crucial for conservation and management strategies (*Gaston, 2000*; *Defeo et al., 2021*). Identifying the main drivers of marine diversity, including spatial and seasonal variations, is critical for establishing a strong baseline for future studies.

Sandy beaches morphodynamics may change at different timescales: along several decades to hundreds of years; from several years to decades; seasonal variability, which repeats on an annual cycle; short-term variability, generally associated with extreme events (*Senechal & Alegría-Arzaburu, 2020*). The interest in understanding variability at different temporal scales (inter- and intra-annual patterns) in marine ecosystems has increased recently (*Blue & Kench, 2017*; *Vos et al., 2019*), boosted by the frequency and intensity of climate change events. It is well known that the southern hemisphere lacks long-term data in tropical and subtropical environments, what represents a risk to the development of a worldwide synthesis regarding the biological diversity and dynamics of marine ecosystems (*Odebrecht et al., 2017*). Accordingly, seasonal variation has been understudied, even though it strongly affects the beach system and the benthic fauna associated (macro and meiofauna) (*Basanta, Sathish Kumar & Karunakar, 2017*; *Senechal & Alegría-Arzaburu, 2020*). Long-term studies are key for accurately assessing changes in the ecosystem, rigorous monitoring of cycles and trends, and an acceptable assessment of the status of living resources (*e.g.*, meiofaunal composition and diversity pattern), so that seasonal variations can be captured and fully understand this ecosystem under constant pressure (*e.g.*, anthropogenic activities and climate change) (*Coppo, 2023*).

Meiofauna is composed by organisms ranging from 42 to 500 µm, comprising at least 22 phyla, and often displaying high abundance and diversity in marine benthic systems (*Higgins & Thiel, 1988*; *Giere, 2009*; *McIntyre, 1969*; *Hakenkamp & Palmer, 2000*). These organisms play crucial ecological roles in marine sediments through nutrient recycling, thus transferring energy and matter into benthic and pelagic trophic food webs and linking different trophic levels (*Giere, 2009*). In benthic marine communities from coastal habitats, spatial-temporal diversity patterns are mostly driven by substrate and oceanographic

parameters (*Blanchette et al., 2008*; *Griffiths et al., 2017*; *Mazzuco et al., 2019*; *Mazzuco, Stelzer & Bernardino, 2020*). It is recognized that sediment grain size, coastal hydrodynamics, and food availability are typical drivers of meiofaunal coastal communities (*Giere, 2009*). In sandy beaches, the distribution and abundance of infaunal benthos are expected to respond to physical factors, such as the swash climate and sediment characteristics (*McLachlan et al., 1993*; *Todaro & Rocha, 2004*; *McLachlan & Brown, 2006*; *Albuquerque et al., 2007*; *Maria et al., 2013*). Wave action also plays an important role on spatial variability (*i.e.*, patchiness) of density and diversity of meiofauna due to the hydrodynamic stress (*Covazzi et al., 2001*). Along the intertidal zone of sandy beaches, temperature and salinity are highly variable and can also influence the distribution and composition of organisms (*Olafsson, 1991*). In tropical areas, seasonal changes are less markedly defined, but meiofaunal organisms show some seasonality, with greater abundance during the warmest/rainy months (*Coull, 1988*; *Albuquerque et al., 2007*). Nevertheless, other studies have shown that biological factors, such as food availability, are also responsible for structuring benthic macrofauna community (*Lastra et al., 2006*; *Rodil, Compton & Lastra, 2012*).

Meiofaunal taxa may have specific adaptations and each taxa respond differently to environmental conditions, due to their differential ability of dispersion, locomotion, nutrition, development, and reproduction (*Curini-Galletti et al., 2012*). A number of studies demonstrated that benthic species richness increases from temperate to tropical sandy beaches for macrofauna (*McLachlan, De Ruyck & Hacking, 1996*; *McLachlan & Dorvlo, 2005*; *Defeo & McLachlan, 2013*) and meiofauna (*Lee & Riveros, 2012*). In tropical humid regions, rainfall may additionally work as a major factor structuring meiofauna diversity in sandy beaches (*Gomes & Rosa-Filho, 2009*; *Venekey, Santos & Fonsêca-Genevois, 2014*; *Baia & Venekey, 2019*). Previous studies have demonstrated that meiofaunal communities respond to warming in aquatic ecosystems, reducing in diversity and abundance (*O'Gorman et al., 2012*; *Gingold, Moens & Rocha-Olivares, 2013*), causing the mortality of dominant species in subtropical environments (*Gingold, Moens & Rocha-Olivares, 2013*), reducing biomass (*Alsterberg, Hulth & Sundbäck, 2011*), and body-size (*Jochum et al., 2012*).

Understanding meiofaunal spatial and temporal diversity patterns is paramount in a scenario of global environmental change, investigating variations over time and predicting future changes is crucial for conservation strategies, management and identifying priority areas for conservation (*Muller-Karger et al., 2017*; *Bax et al., 2019*; *Mazzuco, Stelzer & Bernardino, 2020*; *Strassburg et al., 2020*; *Pittman et al., 2021*). To predict how these assemblages will respond in the future, firstly it is necessary to understand the drivers of local-scale diversity patterns, and how organisms respond to environmental parameters and seasonality (see *Coppo, 2023*). Here, we aimed to assess the meiofaunal diversity in a tropical sandy beach in the SW Atlantic coast and tested whether or not (i) there is seasonal difference in meiofaunal assemblage composition and diversity (phylogenetic and Shannon's diversity) along 1 year, and (ii) the marine seascapes conditions, rainfall, and sediment parameters affect significatively meiofaunal diversity metrics. We addressed the following hypotheses in this study: (i) meiofaunal diversity is higher during warmer

months and its composition changes significatively among seasons along a year at Gramuté beach, and (ii) marine seascapes and sediment characteristics fluctuation along the year influence significantly meiofaunal diversity.

## MATERIALS AND METHODS

### Study area and sampling

The study was carried out at Gramuté, a sandy beach located within a marine protected area in the Eastern Brazilian Marine Ecoregion (Fig. 1A). The region is geomorphically characterized by abrasion terraces of Barreiras Formation from the coast to the inner continental shelf (*Martin et al., 1996*). Gramuté beach is marked by scattered intertidal lateritic reefs (*Mazzuco, Stelzer & Bernardino, 2020*), with the presence of carbonate secreting organism which contributes to the deposition of bioclastic sediment (*Albino & Suguio, 2011*). Additionally, Gramuté beach is marked by strong internal tidal currents, and E-SE wave swells with upwelling events occurring mostly during spring and summer (*Pereira et al., 2005*). This tropical region is marked by dry winters and rainy summers (*Bernardino et al., 2015*), with sea surface temperatures ranging between 21 °C and 27 °C, and salinity ranging from 34.6 to 36 ppt (*Quintana et al., 2015*; *Mazzuco et al., 2019*; *Mazzuco, Stelzer & Bernardino, 2020*). This region has also experienced significant warming in the last 40 years (*Bernardino et al., 2015*; *Mazzuco, Stelzer & Bernardino, 2020*).

The study region is marked by frequent exposure to waves generated mainly by the South Atlantic Subtropical Anticyclone (ASAS), with northeast (NE) swells mainly. Although there is dominance of NE waves throughout the year, in the autumn and winter period the wind regime changes to E-SE, strengthening the presence of waves from these directions (E-SE), with average heights of 1.5 m. During winter, the region is also affected by the passage of frontal systems, making it susceptible to wave action coming from the south-southwest (S-SW) (*Silva, Hansen & Cavalheiro, 1984*).

During a year (December 2019 to November 2020), we monthly collected sediment samples (approximately 200 g each replicate) on three different stations ($n = 9$ sediment samples per month) at Gramuté beach on the low-tide shoreline, spaced 20 m apart (Fig. 1B). Sediment samples were collected manually using sterile, DNA-free corers, over all seasons during the sampling period (Table S1). Additionally, we collected samples for sediment analysis (grain size, total organic matter, carbonate content, and sedimentary organic biopolymers). All samples were transported in thermic bags with ice, and stored at −20 °C until analysis. Field sampling was authorized by the Biodiversity Authorization and Information System of the Brazilian Institute for the Environment and Renewable Natural Resources (SISBIO-IBAMA, sampling license number 24700-1). Total monthly rainfall data for the sampling period (December 2019—November 2020) was obtained from the National Water Resources Information System (SNIRH) portal, made available by the National Water and Sanitation Agency (*Agência Nacional de Águas (ANA) (2023)*; ANA—https://www.snirh.gov.br/hidroweb/), considering the station of Santa Cruz -Litoral (code: 1940002; Lat:−19.9578, Lon:−40.1544), which is approximately 4 Km from Gramuté beach.
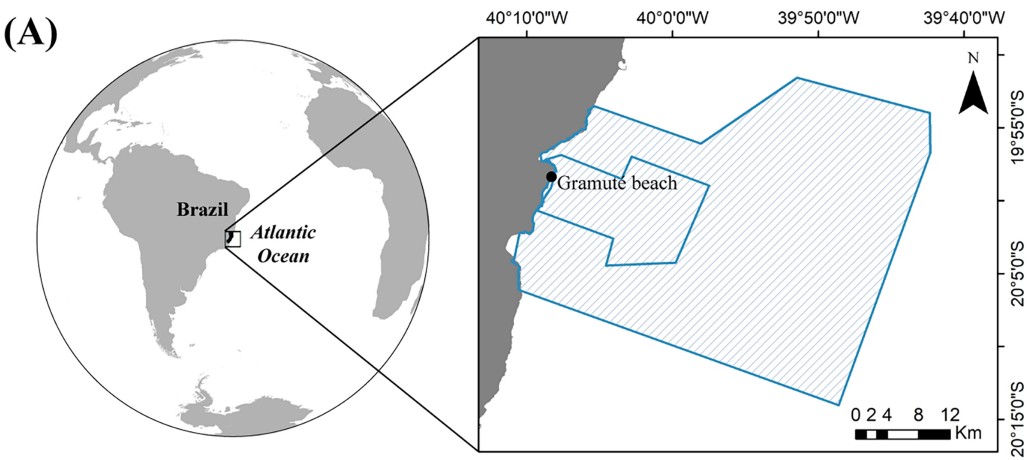

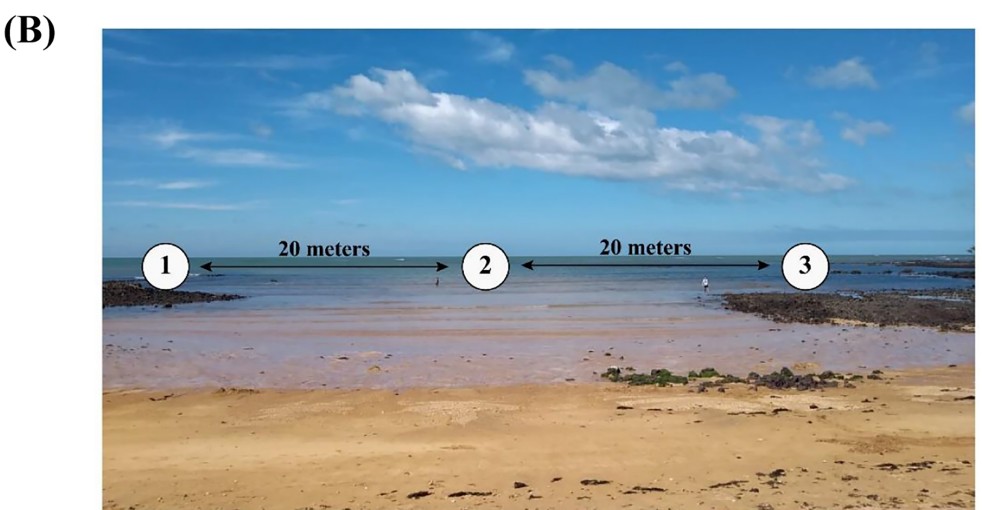

**Figure 1** **Study area location.** (A) Location of Gramuté beach in the SE Brazilian coast, within the marine protected areas Refúgio da Vida Silvestre de Santa Cruz and Área de Proteção Ambiental Costa das Algas (polygon areas); (B) sampling design in Gramuté beach, with sampling stations 20 m apart from each other.

## Sediment analysis

Sediment samples were dried at 60 °C for 48 h before all granulometric analysis. Dried sediment was macerated and sieved in mesh openings of 63 μm to 2 mm in a sieve shaker to determinate the carbonate content by muffle combustion at 550 °C for 4 h with an additional hour at 800 °C. Additionally, we quantified total organic matter (TOM) by weight loss after combustion (500 °C for 3 h) (*Suguio, 1973*).

Sedimentary organic biopolymers (proteins, carbohydrates, and lipids) were analyzed following *Danovaro (2010)*. After extraction with NaOH 0.5 M we determinated total protein (PRT) content according to *Hartree (1972)* as modified by *Rice (1982)* to compensate for phenol interference. For total carbohydrates (CHO) analysis, we followed the protocol from *Gerchacov & Hatcher (1972)*. Total lipids (LIP) were extracted from 1 g of homogenized sediment lyophilized by ultrasonication in 10 ml of chloroform: methanol
(2:0 1 v/v) and analyzed according to *Marsh & Weinstein (1966)*. The concentrations of PRT, CHO, and LIP are presented respectively as bovine serum albumin, glucose, and tripalmitin equivalents. Concentrations of PRT, CHO, and LIP were converter to carbon equivalents following *Fabiano & Danovaro (1994)* using conversion factors of 0.49, 0.40, and 0.75, respectively. The sum of PRT, CHO, and LIP carbon equivalents are presented as biopolymeric carbon (BPC) (*Fabiano, Danovaro & Fraschetti, 1995*). Further, protein to carbohydrate (PRT: CHO) and carbohydrate to lipid (CHO: LIP) ratios were used to assess biochemical degradation processes (*Galois et al., 2000*). All analyzes were performed in triplicate and blanks were carried out for all analysis with pre-combusted sediments at 450 °C and 480 °C for 4 h.

## Seascapes identification

The Marine Biodiversity Observation Network (MBON) Seascapes are a characterization of water masses with particular biogeochemical features obtained from satellite and modeled data that comprises different oceanic parameters (sea surface temperature—SST, sea surface salinity—SSS, absolute dynamic topography—ADT, chromophoric dissolved organic material—CDOM, surface chlorophyll-a–Chl-a, and normalized fluorescent line height—NFLH). These variables are used for a categorization system of 33 water masses that represents different marine scenarios/conditions (*Montes et al., 2020*).

Oceanographic conditions were characterized according to the variation in MBON Seascape Pelagic Habitats Classification (*Kavanaugh et al., 2014, 2016*; *Mazzuco & Bernardino, 2022*) using the database available in the NOAA Coast and Ocean Watch Programs, with monthly frequency on a 5 $Km^2$ grid (*Kavanaugh et al., 2014, 2016*), to characterize the seascapes for the Área de Proteção Ambiental Costa das Algas (~30 km coastline, 465 $Km^2$, Longitude–40.3° to–39.8°, Latitude 20.3° to 19.8°) for the study period (December 2019–November 2020). Additionally, to determine seasonal SST and SSS for the study area, we calculated a weighted average based on the monthly coverage area of each identified MBON marine seascape.

MBON Seascapes are presented as seascape coverage (%), which represents the extent of an area that is encompassed within any of the MBON Marine Seascapes categories. This percentage represents how much of the area of the Área de Proteção Ambiental Costa das Algas (which is a Marine Protected Area) is encompassed with one of the seascape categories. Each MBON Seascape category is defined by a fixed value for each oceanic variable, and the seascape product is generated as monthly and 8-day composites at 5 Km spatial resolution. In this manuscript we used the seasonal mean, calculated as the mean of the monthly seascape coverage for all 3 months per season.

## DNA extraction and sequencing

Prior to DNA extraction, sediment samples were elutriated using sieves of 45 μm mash in an attempt to increase the meiofaunal abundance and enrich metazoan DNA recovery as suggested by *Brannock & Halanych (2015)*.

A total of 1 L flasks were filled with 950 mL of filtered seawater and sediment samples were added to it, then homogenized and let to sit (for approximately 30 s) before decanting

the liquid over the 45 μm sieve. This procedure was repeated 10 times for each sediment sample. The sediment retained on the sieve was transferred to 50 mL Falcon tubes and centrifuged at room temperature for 3 min at 1,342 × g in an Eppendorf Centrifuge 5430, then the sample volumes were standardized to 20 mL. We mixed the samples in the Falcon tubes, then aliquots of 1 mL (two tubes per sample) were stored at −20 °C (*Brannock & Halanych, 2015*). All glassware was cleaned and autoclaved between samples to avoid cross contamination between samples, and sieves were sterilized by soaking for 45 min in 10% sodium metabisulfite solution (*Creer et al., 2010*; *Brannock & Halanych, 2015*). DNA was extracted from 1 mL aliquots using the PowerSoil DNA® (Qiagen) kit following the manufacturer's instructions. DNA integrity was verified in 1% agarose gel, and purity using NanoDrop One spectrophotometer (Thermo Fisher Scientific Inc., Waltham, MA, USA). We measured DNA concentration using the Qubit® 4 Fluorometer (Qubit™ 1X dsDNA HS Assay Kit—Life Technologies-Invitrogen, Carlsbad, CA, USA). Blank samples were carried in triplicate for each step before metabarcoding sequencing (sediment elutriation, DNA extraction, and integrity, purity, and concentration checking).

DNA samples extracted from the same sediment sampling station, for each month separately, were combined into a single pool, totaling three samples per month (nine samples per season, totaling 36 samples). PCR, library preparation, and sequencing were conducted by ©NGS Genomic Solutions (Piracicaba, SP, Brazil). Metabarcoding sequencing was performed using the MiSeq Illumina platform (2 × 250 bp, with a coverage of 100,000 paired-end reads per sample), sequencing the V9 hypervariable region from 18S SSU rRNA gene using the primers Euk_1391 forward (GTACACACCGCCCGTC) and EukBr reverse (TGATCCTTCTGCAGGTTCACCTAC) (*Medlin et al., 1988*; *Amaral-Zettler et al., 2008*; *Stoeck et al., 2010*), generating amplicons that could vary in size (mean 260 ± 50 bp), once the reverse primer doesn't have its exact position conserved as the forward one.

## Bioinformatic pipeline

Bioinformatic analysis were conducted using an AMD Ryzen 1950× Crucial 64 GB (16 × 4) DDR4 2,666 MHz computer. We used the QIIME2 2022.8 software to identify sequences with the demultiplexed raw paired-end reads (*Bolyen et al., 2019*). Firstly, we imported FastQ files as QIIME2 artifacts, then denoised them *via* DADA2 (*Callahan et al., 2016*) using the *denoise-paired* plugin, and removed low-quality bases and primer sequences.

The taxonomic assignment of amplicon sequence variants (ASV) generated by the DADA2 plugin (p-trim = 10, p-trunc = 160, and mean phred score = 39 ± 1; Table S2) was performed using the machine learning Python library scikit-learn (*Pedregosa et al., 2011*). A pre-trained Naïve Bayes classifier trained on the Silva 138 database (*Quast et al., 2013*) clustered at 99% similarity was used to identify taxonomically the DNA sequences. Due to the differences on the number of identified sequences, the dataset was normalized to allow analysis and comparisons under equal sampling depth. We used the spring dataset minimum sampling depth (1,384 reads) and resampled each sample to the same depth. This normalized dataset was used to calculate all diversity metrics. We performed rarefaction curves for all four sampled seasons (summer, winter, spring, and autumn) with

the ASVs. We calculated the Faith's Phylogenetic Diversity (PD) for each sample using the *diversity core-metrics-phylogenetic* pipeline from QIIME2. The PD was calculated based on phylogenetic trees generated using the *align-to-tree-mafft-fasttree* pipeline from the *q2-phylogeny* plugin from QIIME2. Shannon diversity was calculated using the *qiime diversity alpha pipeline* and setting the p-metric parameter to "Shannon". Raw sequences are available online on NCBI (SRR24675047) and on Brazilian Biodiversity Information System (SiBBr; *Bernardino & Coppo, 2024*).

## Statistical analysis

In our statistical analyses, only meiofaunal metazoan sequences used. The dataset was subset to contain only metazoan sequences based on taxonomic annotations using the *qiime taxa filter-table* function from the QIIME2 *q2-taxa* plugin. Here we considered all the exclusively meiofaunal phyla (Gnathostomulida, Kinorhyncha, Loricifera, Gastrotricha, and Tardigrada) and other metazoans that can be meiofaunal-sized during their life cycle (temporary meiofaunal taxa) (*Higgins & Thiel, 1988*; *Giere, 2009*). After this filtering, the final dataset contained 10 phyla (Annelida, Cnidaria, Crustacea, Echinodermata, Gastrotricha, Mollusca, Nematoda, Nemertea, Platyhleminthes, and Rotifera) as previously implemented in other studies (*Brannock & Halanych, 2015*; *Bernardino et al., 2019*; *Fais et al., 2020*; *Bellisario et al., 2021*; *Castro et al., 2021*; *Coppo et al., 2023*). Permutational analysis of variance (PERMANOVA; *Anderson, Gorley & Clarke, 2008*) was performed to compare environmental variables (rainfall, temperature, salinity, carbonate content, grain size, total organic matter, and biopolymeric composition), seascape coverage (the extent of an area that is encompassed within any of the MBON Marine Seascapes categories), and meiofaunal data (diversity metrics— Shannnon's diversity index, phylogenetic diversity, and abundance of sequence reads) among sampled seasons (summer, autumn, winter, and spring) and sampled stations at Gramuté beach. Bonferroni corrections was used to adjust *p*-values for pairwise comparisons (*Bonferroni, 1936*). A canonical analysis of principal coordinates (CAP; *Anderson & Willis, 2003*) was performed with environmental variables (rainfall and sediment variables) and the meiofaunal assemblage composition at Phylum level (square-root transformed). Additionally, a similarity percentage routine (SIMPER; *Clarke, 1993*) was applied to define the taxa that most contributed to the dissimilarity among seasons, based on a Bray-Curtis dissimilarity matrix. A multiple linear regression was fit using Shannon's diversity Index and phylogenetic diversity as response variables, and the assessed environmental variables as predictive variables. After testing for multicollinearity among variables, we removed carbohydrate content (CHO) which was highly correlated to carbohydrate-to-lipids ratio (CHO:LIP) and biopolymeric carbon (BPC). Normality tests were run on model's residuals through QQ-plots and Shapiro-Wilk normality tests. After obtaining the multiple linear regression values, we used the Akaike Information Criterion (AIC), through a stepwise backward model configuration and the final model was chosen based on the lowest AIC value (*Akaike, 1978*). Significative differences were defined at $p < 0.05$. All graphical and analytical procedures were performed in the R environment (*R Core Team, 2022*).

## RESULTS

### Environmental conditions and seascape coverage

Significant seasonal variability was observed at Gramuté beach during the period studied (PERMANOVA, df = 3; Pseudo-F = 6.916; $p$ = 0.001; Table S3), with higher LIP content on autumn (Fig. 2; Table S3) and lower on spring (Fig. 2; Table S3), meanwhile the CHO:LIP ratio was higher on winter than on other seasons (Fig. 2; Table S3). Total rainfall ranged from 80.2 ± 38.4 mm in summer to 193.0 ± 42.2 mm in autumn (Fig. 2). The sediment is completely composed of sand, mainly by medium and coarse sand, with carbonate content ranging from 26 ± 10% during spring to 55 ± 10% in winter (Fig. 2). Total organic matter (TOM) had its lower concentration in summer (8.6 ± 3.8%), and higher in spring (10.4 ± 7.9%) (Fig. 2; Table S3). The protein fraction of the organic matter content in the sediment ranged from 48.2 ± 21.6 mg/g in autumn to 96.9 ± 14.8 mg/g in summer (Fig. 2; Table S3), while the carbohydrate fraction ranged from 997.1 ± 193.5 mg/g in autumn to 2,102.0 ± 1,435.0 mg/g in winter (Fig. 2; Table S3). The labile fraction of the organic matter, which is represented by the BPC, was similar among seasons (Fig. 2; Table S3).

Overall, the Seascapes categories in this region were characterized by high sea surface temperature (SST > 20.9 °C), high sea surface salinity (SSS > 33.6 psu) and calm waters (absolute dynamic topography-ADT ranging from 0.51 to 0.83 m). The seascapes had wide ranges in dissolved organic matter (CDOM; 0.00 to 0.07 $m^{-1}$), chlorophyll-a concentration (CHLA; 0.07 to 2.09 $mg.m^{-3}$), and fluorescence (NFLH; 0.02 to 0.24 $W.m^{-2}.um^{-2}sr^{-1}$) (Fig. 3). We observed changes in the frequency of seascapes in the studied area along the year (PERMANOVA, df = 3; Pseudo-F = 8.014; $p$ = 0.001; Table 1). Seascapes Tropical Seas (class 15–38.4% of area coverage during sampling period), Subtropical Gyre Transition (class 5–19.0% of area coverage during sampling period), Subtropical Gyre Mesoscale Influenced (class 13–18.3% of area coverage during sampling period), and Warm, Blooms, High Nutrients (class 21–12.4% of area coverage during sampling period) were the most frequent, with more than 80% of area coverage during the study period (Fig. 3).

Water masses at Gramuté beach during summer (Dec–Feb), autumn (Mar–May) and winter (Jun–Aug) were dominated by the Seascape Tropical Seas (class 15), which is characterized by high temperatures (25.4 °C) and salinity (35.4 psu), and covered with 40.9% (summer), 43.1% (autumn), and 45.1% (winter) of the study area (Fig. 3). During spring, the dominance of seascapes at Gramuté changed due to an intrusion of subtropical a water mass (class 13–42.7% of area coverage; Fig. 3), characterized by significant lower temperature (23.5 °C) and higher salinity (35.9 psu).

### Meiofaunal assemblage

A total of 9,692 sequences from meiofaunal taxa were identified in the dataset. We did not observe significative differences in meiofauna composition among sampled stations (PERMANOVA, df = 2; Pseudo-F = 0.963; $p$ = 0.491; Table 2). Nonetheless, there were significant seasonal variations, with winter differing from all other seasons (PERMANOVA, df = 3; Pseudo-F = 2.307; $p$ = 0.002; Table 2; Fig. 4). During the summer

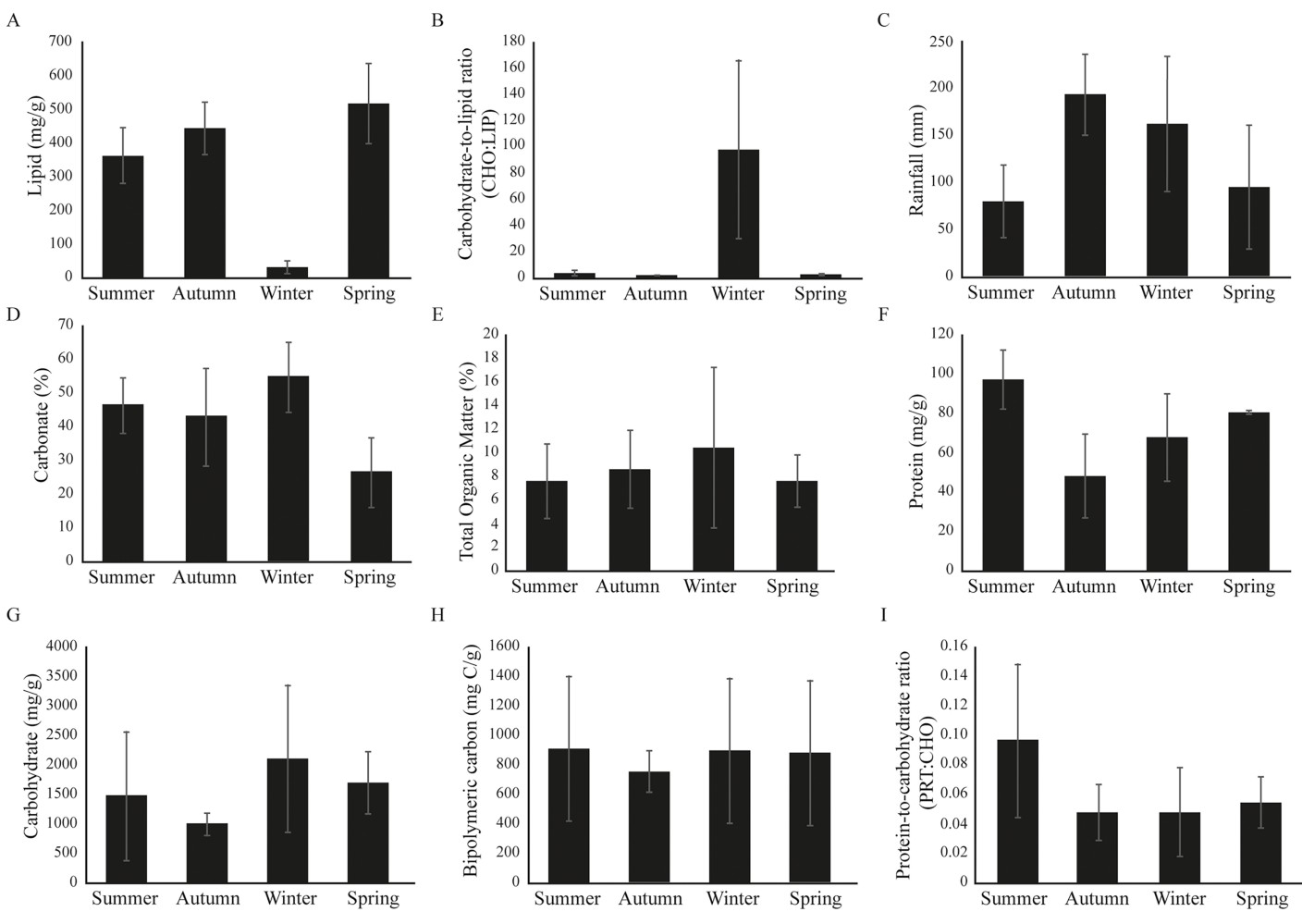

**Figure 2 (A–I) Environmental variables barplots.** Environmental variables (mean ± SD) from Gramuté beach, SE Brazil, during all seasons (summer, autumn, winter, and spring).

(35% and 40% of reads), autumn (43% and 34% of reads), and spring (59% and 27% of reads), Crustacea and Annelida were the most prevalent taxa. The two taxa with the highest abundance throughout the winter were Crustacea (57% of reads) and Nematoda (17% of reads) (Fig. 4). Nemertea was not detected during autumn, Gastrotricha was not detected in spring, and Rotifera was not detected in neither. Only 11 taxa (*e.g.*, Harpacticoida, Podocopida, and Chromadorea) were detected on all seasons (Table S4), meanwhile 14 taxa were detected only on one sampled season (Table S4).

Fewer meiofaunal taxa were detected in spring than all other seasons (Fig. 5), influencing on significant seasonal differences on diversity patterns in Gramuté beach. Lower phylogenetic diversity was registered in spring (9.23 ± 1.88) and in autumn (11.88 ± 1.82) than in summer (17.93 ± 3.11) and winter (19.37 ± 4.85) (PERMANOVA; Pseudo-F = 18.863; df = 3; $p < 0.001$; Table 3). Similarly, Shannon's diversity was 1.7-fold, 2.0-fold, and 1.9-fold lower in spring than in autumn, summer, and winter, respectively (PERMANOVA; Pseudo-F = 13.129; df = 3; $p < 0.001$; Table 4).

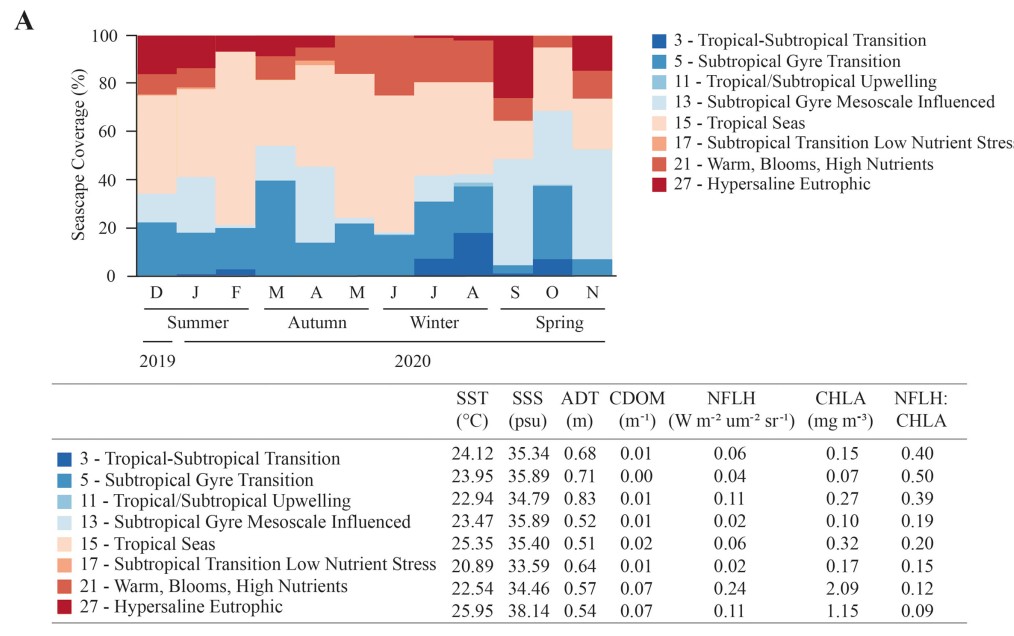

| | SST (°C) | SSS (psu) | ADT (m) | CDOM (m⁻¹) | NFLH (W m⁻² um⁻² sr⁻¹) | CHLA (mg m⁻³) | NFLH: CHLA |
|---|---|---|---|---|---|---|---|
| 3 - Tropical-Subtropical Transition | 24.12 | 35.34 | 0.68 | 0.01 | 0.06 | 0.15 | 0.40 |
| 5 - Subtropical Gyre Transition | 23.95 | 35.89 | 0.71 | 0.00 | 0.04 | 0.07 | 0.50 |
| 11 - Tropical/Subtropical Upwelling | 22.94 | 34.79 | 0.83 | 0.01 | 0.11 | 0.27 | 0.39 |
| 13 - Subtropical Gyre Mesoscale Influenced | 23.47 | 35.89 | 0.52 | 0.01 | 0.02 | 0.10 | 0.19 |
| 15 - Tropical Seas | 25.35 | 35.40 | 0.51 | 0.02 | 0.06 | 0.32 | 0.20 |
| 17 - Subtropical Transition Low Nutrient Stress | 20.89 | 33.59 | 0.64 | 0.01 | 0.02 | 0.17 | 0.15 |
| 21 - Warm, Blooms, High Nutrients | 22.54 | 34.46 | 0.57 | 0.07 | 0.24 | 2.09 | 0.12 |
| 27 - Hypersaline Eutrophic | 25.95 | 38.14 | 0.54 | 0.07 | 0.11 | 1.15 | 0.09 |

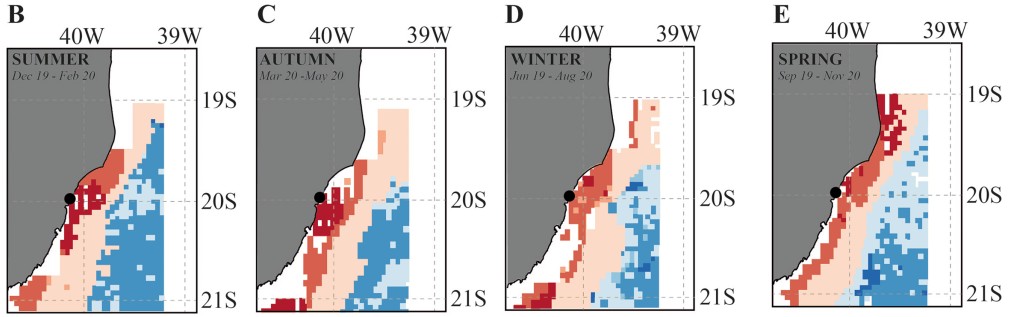

**Figure 3 Seascape coverage.** Monthly (A) and seasonal (B–E) variation in Seascapes coverage (%) between December 2019 to November 2020 in Gramuté beach, SE Brazil. Mean oceanographic values from oceanographic variables that identify each MBON Seascape water mass (class). SST-sea surface temperature, SSS-sea surface salinity, ADT-absolute dynamic topography, CDOM-chromophoric dissolved organic material, CHLA-chlorophyll-a, NFLH-normalized fluorescent line height.

**Table 1 PERMANOVA results.**

| Source | df | SS | MS | Pseudo-F | p |
|---|---|---|---|---|---|
| Season | 3 | 120.1 | 40.0 | 8.014 | **0.001** |
| Residual | 32 | 159.9 | 5.00 | | |
| Total | 35 | 280.0 | | | |
| **Pair-wise tests** | | | | | |

| Groups | t | p |
|---|---|---|
| Summer × Autumn | 1.478 | 0.106 |
| Summer × Winter | 3.467 | **0.001** |
| Summer × Spring | 3.003 | **0.002** |

(Continued)

**Pair-wise tests**

| Groups | t | p |
|---|---|---|
| Autumn × Winter | 2.461 | **0.001** |
| Autumn × Spring | 2.589 | **0.002** |
| Winter × Spring | 3.573 | **0.001** |

Note:
Permutational multivariate analysis of variance results from MBON Seascapes coverage at local scale (~30 km coastline, 465 $Km^2$) at SE Brazil, during all seasons (summer, autumn, winter, and spring). Significative results are considered when $p < 0.05$, and are presented in bold. DF, Degrees of Freedom; SS, sum of squares; MS, mean of squares.

**Table 2 PERMANOVA results.**

| Source | df | SS | MS | Pseudo-F | p |
|---|---|---|---|---|---|
| Season | 3 | 1,1842.0 | 3,947.5 | 2.307 | **0.002** |
| Station | 2 | 3,296.2 | 1,648.1 | 0.963 | 0.491 |
| Season × Station | 6 | 9,315.7 | 1,552.6 | 0.907 | 0.657 |
| Residual | 24 | 41,070.0 | 1,711.3 | | |
| Total | 35 | 65,525.0 | | | |

**Pair-wise tests**

| Groups | t | p |
|---|---|---|
| Autumn × Spring | 1.356 | **0.044** |
| Autumn × Summer | 1.074 | 0.312 |
| Autumn × Winter | 1.403 | 0.062 |
| Spring × Summer | 1.950 | **0.001** |
| Spring × Winter | 1.880 | **0.001** |
| Summer × Winter | 0.988 | 0.488 |
| Station 1 × Station 2 | 1.198 | 0.214 |
| Station 1 × Station 3 | 0.852 | 0.698 |
| Station 2 × Station 3 | 0.922 | 0.575 |
| Autumn, Station 1 × Station 2 | 0.958 | 0.611 |
| Autumn, Station 1 × Station 3 | 0.322 | 1.000 |
| Autumn, Station 2 × Station 3 | 0.946 | 0.502 |
| Spring, Station 1 × Station 2 | 1.015 | 0.497 |
| Spring, Station 1 × Station 3 | 1.349 | 0.205 |
| Spring, Station 2 × Station 3 | 1.203 | 0.396 |
| Summer, Station 1 × Station 2 | 0.792 | 0.606 |
| Summer, Station 1 × Station 3 | 0.365 | 0.885 |
| Summer, Station 2 × Station 3 | 0.703 | 0.906 |
| Winter, Station 1 × Station 2 | 1.113 | 0.429 |
| Winter, Station 1 × Station 3 | 0.776 | 0.785 |
| Winter, Station 2 × Station 3 | 0.652 | 0.886 |

Note:
Permutational multivariate analysis of variance results from meiofaunal composition at Gramuté beach, SE Brazil, during all seasons (summer, autumn, winter, and spring) and sampled stations. Significative results are considered when $p < 0.05$, and are presented in bold. Df, Degrees of Freedom; SS, Sum of squares; MS, Mean of squares.

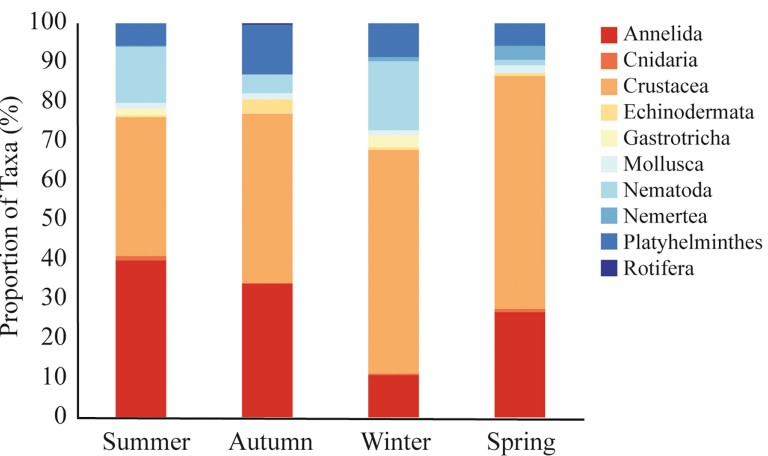

**Figure 4 Meiofaunal assemblage composition.** Meiofaunal taxa proportion (%) at Phyllum level in all sampled seasons at Gramuté beach, SE Brazil.

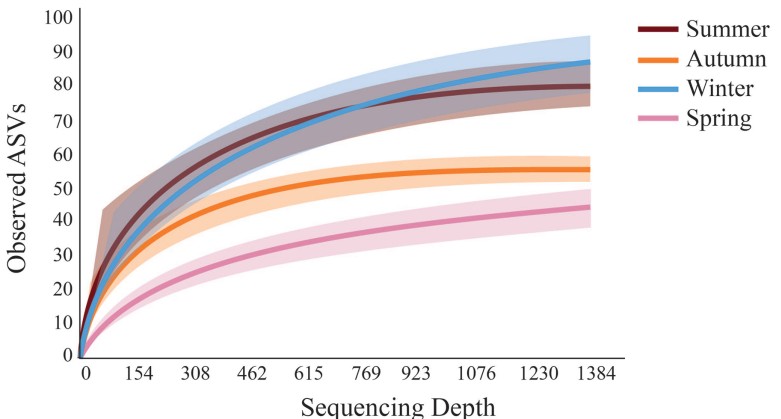

**Figure 5 Rarefaction curves obtained from sediment samples metabarcoding collected at Gramuté beach, SE Brazil, during all seasons on a 1-year sampling.** Solid lines represent a mean of observed ASVs at each sampling depth, and the shaded area represents the standard deviation.

Meiofaunal assemblages differed significantly among the seasons in Gramuté beach (PERMANOVA, df = 3; Pseudo-F = 2.353; $p$ = 0.001; Table 2; Table S4). Dissimilarity levels ranged from 49.7% (between winter and summer) to 68.6% (between autumn and summer). SIMPER analysis revealed that Annelida (ranging from 16.5% to 28.3%; Table S5), Crustacea (ranging from 21.8% to 26.7%; Table S5) and Nematoda (ranging from 13.9% to 21.8%; Table S5) were the taxa that most contributed to the differences among all seasons. Platyhelminthes contributed 15.4% to the total dissimilarity of 49.5% between autumn and spring (Table S5). Annelids, crustaceans, and nematodes were more abundant in summer and winter. Furthermore, these taxa were highly associated with higher organic matter content and quality (total organic matter content, biopolymeric carbon, protein content, and protein-to-carbohydrate ratio; Fig. 6).

**Table 3 PERMANOVA results.**

| Source | df | SS | MS | Pseudo-F | p |
|--------|----|----|-----|----------|---|
| Season | 3 | 1,545.9 | 515.3 | 18.863 | **0.001** |
| Residual | 32 | 573.69 | 27.3 | | |
| Total | 35 | 2,119.6 | | | |

**Pair-wise tests**

| Groups | t | p |
|--------|---|---|
| Summer × Autumn | 3.699 | **0.022** |
| Summer × Winter | 0.508 | 0.657 |
| Summer × Spring | 6.186 | **0.001** |
| Autumn × Winter | 3.303 | **0.017** |
| Autumn × Spring | 2.219 | 0.054 |
| Winter × Spring | 6.012 | **0.001** |

Note:
Permutational multivariate analysis of variance results from meiofaunal phylogenetic diversity at Gramuté beach, SE Brazil, during all seasons (summer, autumn, winter, and spring). Significative results are considered when $p < 0.05$, and are presented in bold. Df, Degrees of Freedom; SS, Sum of squares; MS, Mean of squares.

**Table 4 PERMANOVA results.**

| Source | df | SS | MS | Pseudo-F | p |
|--------|----|----|-----|----------|---|
| Season | 3 | 1,954.2 | 651.4 | 13.129 | **0.001** |
| Residual | 32 | 1,587.6 | 49.6 | | |
| Total | 35 | 3,541.8 | | | |

**Pair-wise tests**

| Groups | t | p |
|--------|---|---|
| Summer × Autumn | 2.598 | **0.026** |
| Summer × Winter | 0.719 | 0.544 |
| Summer × Spring | 4.906 | **0.001** |
| Autumn × Winter | 0.899 | 0.412 |
| Autumn × Spring | 3.861 | **0.001** |
| Winter × Spring | 3.806 | **0.003** |

Note:
Permutational multivariate analysis of variance results from meiofaunal Shannon's Diversity index at Gramuté beach, SE Brazil, during all seasons (summer, autumn, winter, and spring). Significative results are considered when $p < 0.05$, and are presented in bold. Df, Degrees of Freedom; SS, Sum of squares; MS, Mean of squares.

Protein content (PRT), lipid content (LIP), biopolymeric carbon (BPC), and carbohydrate-to-lipids ratio (CHO:LIP) composed a significant model of variables likely to drive meiofauna diversity (Shannon's diversity Index and phylogenetic diversity) at Gramuté beach (Adjusted $R^2 = 0.602$; F = 7.319; $p < 0.001$; Table 5). We observe significant positive relationship between meiofaunal diversity (Shannon's diversity and phylogenetic diversity) and lipid content (LIP) (t = 2.513; $p = 0.018$; Table 5). Biopolymeric carbon content—BPC showed a significant negative relationship to diversity (t = −2.584; $p = 0.015$; Table 5), as well as protein content (PRT), although it was not significative (t = −1.719; $p = 0.096$; Table 5).

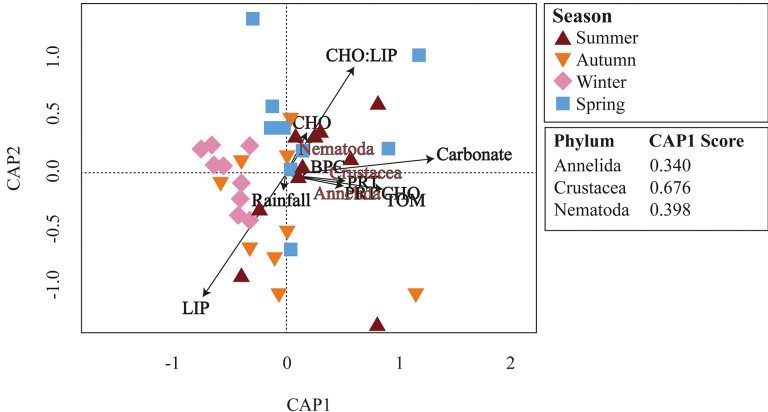

**Figure 6 CAP ordination.** Canonical analysis of principal coordinates (CAP) of main meiofaunal phyla and environmental variables (rainfall, carbonate content, organic matter, and biopolymers) at Gramuté beach, SE Brazil, during all seasons.

**Table 5 Linear model results.** Linear model statistical values from relation between meiofaunal diversity (Shannon's diversity index and phylogenetic diversity) and environmental variables (biopolymeric carbon, carbohydrate-to-lipid ratio, lipid content, and protein content) collected in Gramuté beach, SE Brazil, during all seasons (summer, autumn, winter, and spring). This model was chosen based on the lowest AIC. Significative results are considered when $p < 0.05$.

|  | Estimate | Standard error | t | p |
|---|---|---|---|---|
| Intercept | −8.429 | 7.993 | −1.055 | 0.300 |
| Biopolymeric Carbon (BPC) | −0.002 | 0.001 | −2.584 | **0.015** |
| Carbohydrate-to-lipid (CHO:LIP) | 0.015 | 0.009 | −1.530 | 0.137 |
| Lipid (LIP) | 0.006 | 0.002 | 2.513 | **0.018** |
| Protein (PRT) | −0.015 | 0.009 | −1.719 | 0.096 |

# DISCUSSION

This study investigated the influence of seasonal changes on meiofaunal diversity in a sandy beach in the SW Atlantic, advancing the current knowledge on meiofaunal assemblages, and understanding which factors act as main drivers of meiofaunal diversity in a local-scale. Traditionally, benthic sandy beach diversity is expected to be controlled by physical factors (*e.g.*, grain size and tidal action) (*McLachlan et al., 1993*; *Todaro & Rocha, 2004*; *McLachlan & Brown, 2006*; *Albuquerque et al., 2007*; *Di Domenico, Da Cunha Lana & Garraffoni, 2009*; *Maria et al., 2013*; *Maria, Wandeness & Esteves, 2016*). However, previous studies suggested that the sandy beach benthic macrofauna is not structured by a unique physical factor, but by a complex set of drivers also including biological factors (*e.g.*, food availability) (*Lastra et al., 2006*; *Rodil, Compton & Lastra, 2012*). Similarly, *Corte et al. (2022)* reinforced the importance of food supply for sandy beach benthic diversity in other SE Brazil beaches. Our metabarcoding and environmental data demonstrate that this may be similar for sandy beach meiofauna, as biopolymeric carbon (labile fraction of organic matter) and lipid content were the main drivers of meiofaunal diversity at Gramuté beach.

Although seasonal variations of environmental parameters are less markedly defined in tropical environments (*Coull, 1988*; *Albuquerque et al., 2007*), we observed variation on sediment biopolymers content and biopolymeric carbon. Lower quality organic matter was higher during spring and autumn (higher carbohydrate content, lower PRT:CHO, and high CHO:LIP), due to accumulation of aged and degraded organic matter (*Danovaro, Fabiano & Della Croce, 1993*; *Joseph et al., 2008*). The multiple regression fitted showed that food availability (BPC content; *Danovaro, Fabiano & Della Croce, 1993*; *Fabiano, Danovaro & Fraschetti, 1995*) and organic matter quality (PRT, LIP, and CHO:LIP) were the main variables that drive meiofaunal diversity in this study. This result may suggest that the accumulation of aged and degraded organic matter (higher CHO:LIP) is associated to lower meiofaunal diversity, similar to what was reported by *Venturini et al. (2012)*. Similarly, *Cisneros et al. (2011)* observed seasonal changes in organic matter content and nutrients associated to differences in benthic abundance and diversity at a tropical sandy beach. Surprisingly, high-quality organic matter (PRT) that is usually first consumed (*Joseph et al., 2008*) was also negatively associated to meiofaunal diversity in this study. It may be caused by biological interaction among taxa, such as Crustacea and Annelida, which are the main components of the meiofaunal assemblage at Gramuté beach, also known to be strong predators that competitively may suppress the overall diversity. Similar studies supporting the relation between food supply and benthic diversity have been previously reported (*Antón et al., 2011*; *Neto, Bernardino & Netto, 2021*).

Oceanographic conditions and marine seascapes in the study area were mainly characterized by high temperatures, salinity, and nutrients during summer, winter, and autumn. However, this tropical water mass is substituted by an intrusion of a subtropical water mass on spring. This corroborates with the findings of *Silva, Hansen & Cavalheiro, (1984)* and *Perenco (2009)*. This seascape dynamic is influenced by the Brazil Current occurring outside the continental shelf; drift currents generated by winds on the platform up to the wave breaking zone, and currents generated by waves. A similar seascape seasonal pattern was observed by *Mazzuco & Bernardino (2022)* who reported seasonality on benthic recruitment, with higher abundance correlated to warmer water masses and high nutrient content. Changes in marine seascapes are associated to benthic-pelagic interactions and oceanic processes (*Ehrnsten et al., 2019*) that may impact larval supply and recruitment, and consequently the coastal ecosystems biodiversity once many marine animals that live in sandy beach environments have a life cycle with larval/juvenile stage (*Caley et al., 1996*; *Strathmann et al., 2002*; *Mazzuco & Bernardino, 2022*).

Overall, meiofaunal assemblage at Gramuté beach was mainly dominated by Crustacea and Annelida (46% and 28% of reads), with Nematoda representing only 12% of the meiofauna over the year. Nematodes often dominate meiofauna in benthic habitats, with high diversity in the full range of beach types (*Maria et al., 2016*), for example representing 50–90% of the total individuals in medium to fine sandy sediments (*Coull, 1988*; *Giere, 2009*; *Merckx et al., 2009*) but also showing high diversity in coarse sand beaches (*Gheskiere et al., 2005*). However, crustaceans and nematodes become more representative during the winter (57% and 17% of sequence reads), while annelids are less (11% of sequence reads). During the summer annelids and crustacean were more representative

(40% and 35% of sequence reads), and nematodes represented 14% of the sequence reads, differently from what was indicated by *Coull (1988)* for temperate regions.

Crustacea and Annelida are typically macrofaunal groups, however, most crustacean reads observed are from Harpacticoida (21% of the meiofaunal sequences) and Podocopida (13% of the meiofaunal sequences). Harpacticoids are usually one of the most abundant meiofaunal metazoan in sediment samples, that have been reported to represent 35% of the meiofaunal assemblage in tropical beaches, and Podocopida is composed by many marine benthic forms that are meiobenthic size. Some Annelids (Polychaeta) are meiobenthic size as adults, and many polychates have a juvenile phase in meiobenthic size range (temporary meiofauna; *McIntyre, 1969*), such as juvenile Syllidae and Capitellidae (*Giere, 2009*). Most Annelida sequence reads observed in this study are identified as capitellids (16% of the meiofaunal sequences).

Meiofaunal structure, Shannon's diversity index, and phylogenetic diversity were significatively different among seasons, showing that these biological parameters are influenced by seasonal variability, as observed by previous studies in different sandy beaches around the world (*McLachlan & Brown, 2006*; *Baia & Venekey, 2019*; *Baldrighi et al., 2019*). Shannon's diversity was lower during spring, but with no significant differences between summer, autumn, and spring. Phylogenetic diversity was higher in summer and winter than in autumn and spring. These shows that, at Gramuté beach, meiofaunal diversity changes seasonally but is not significantly influenced by rainfall (at Phylum level), although it is known to be important for assemblage structuring as observed by *Gomes & Rosa-Filho (2009)* and *Venekey, Santos & Fonsêca-Genevois (2014)* for nematofauna structure on tropical region. However, it did not play a key role on meiofaunal diversity at our study area (which may be influenced by the taxonomic resolution).

This metabarcoding assessment is the first molecular record of benthic animals registered for this region, and can be used as a baseline dataset for future research. We understand that metabarcoding approaches are influenced by PCR errors, primer biases, and sequence length (*Adams et al., 2019*; *Beng & Corlett, 2020*). Also, the taxonomic identification refinement obtained using DNA-based techniques are directly influenced by the lack of DNA sequences broadly representing meiofauna (*Steyaert et al., 2020*; *Castro et al., 2021*), incomplete DNA-barcodes deposited in molecular databases, and methodological practices (*Cahill et al., 2018*; *Pawlowski et al., 2022*; *Keck et al., 2022*; *Willassen et al., 2022*). Therefore, it is important to emphasize the necessity to follow environmental DNA standards, particularly in sample collection, DNA extraction, genetic marker selection, and reduce or avoid false and negative detection (*Shu, Ludwig & Peng, 2020*). Additionally, metagenomic data should follow the FAIR principles, being findable, accessible, interoperable, and reusable (*ten Hoopen et al., 2017*).

Our study revealed a distinct meiofaunal structure with seasonal influences on diversity at a tropical beach within a Marine Protected Area (MPA), which needs to be considered as a priority area for conservation and management. Understanding diversity patterns and

how it changes seasonally at a local-scale (as well as regional and global) is a key factor for conservation strategies, and associated to it, identifying priority areas for conservation (*Strassburg et al., 2020*; *Pittman et al., 2021*). The number of studies focused on the ecology of sandy beaches in Brazil has increased significantly over the last 10 years. However, the scientific knowledge acquired is not yet sufficient to effectively protect this ecosystem. Management and governance strategies and programs are also critical for identifying and protecting priority areas to maintain diversity and ecosystem services (*Harris & Defeo, 2022*). Open access information and data (published in open-access databases, such as OBIS and GBIF) should be accompanied by management and participatory decision-making process to allow a sustainable management and protection of ecosystem services and benefits from sandy beach ecosystems (*Fanini, Defeo & Elliott, 2020*).

Our findings also highlight the importance of using integrative approaches, including sedimentary variables associated to climatic and water parameters, such as marine seascapes. Additionally, we highlight the importance of long-term studies to understand how meiofaunal assemblages may vary temporally in tropical regions. Long-term ecological studies in sandy beaches are still scarce in Brazil, even with the introduction of monitoring protocols (*e.g.*, MBON Pole-to-Pole and ReBentos) (*Corte et al., 2023*). The scarcity of long-term ecological studies contributes to the limited knowledge on the ecological role of meiofaunal species in sandy beach habitats (*Fanini, Defeo & Elliott, 2020*; *Corte et al., 2023*). Several benthic ecological processes (*e.g.*, recruitment, zonation, intra- and inter-specific interactions) may change during long-term temporal scale (years to decades), and can only be detected and understood with long-term monitoring studies (*Turra et al., 2014*).

## CONCLUSION

Our data help to advance the current knowledge on meiofaunal assemblages, understanding which factors act as main drivers of meiofaunal diversity in a local-scale. We observed seasonal influence on meiofaunal diversity (phylogenetic and Shannon's diversity) at Gramuté beach, where the marine seascape is characterized by high temperatures, high salinity, and calm water masses with high nutrient supply. Higher abundance of reads and diversity were observed during the warmer months of the year (summer), associated to changes in food supply. Abundance of reads and meiofaunal diversity lowered with the intrusion of a subtropical water mass. Additionally, our results reveal that meiofaunal diversity is drive by a complex set of variables, also including biological variables like food supply (biopolymeric carbon–labile fraction of organic matter) and organic matter quality (protein content, lipid content, and carbohydrate-to-lipid ratio), and may be influenced by ecological interactions among taxa. We highlight the necessity of long-term monitoring programs to continue understanding which environmental factors are the main drivers of marine diversity, including spatial and seasonal variations, and how marine benthic organisms will respond to future warmer environmental scenarios.

### Funding

This work was funded by PELD, PRONEM, PROFIX, and Universal grants from Fundação de Amparo à Pesquisa e Inovação do Espirito Santo (790548684/2017, 84532092/2018, 15/2022, 494/2021). Gabriel C. Coppo received a doctorate scholarship from Coordenação de Aperfeiçoamento de Pessoal em Nível Superior CAPES, and a postdoctoral fellowship from Fundação de Amparo à Pesquisa e Inovação do Espirito Santo. The funders had no role in study design, data collection and analysis, decision to publish, or preparation of the manuscript.

### Grant Disclosures

The following grant information was disclosed by the authors:
PELD, PRONEM, PROFIX.
Universal grants from Fundação de Amparo à Pesquisa e Inovação do Espirito Santo: 790548684/2017, 84532092/2018, 15/2022 and 494/2021.
Coordenação de Aperfeiçoamento de Pessoal em Nível Superior CAPES.
Fundação de Amparo à Pesquisa e Inovação do Espirito Santo.

### Competing Interests

Angelo F Bernardino is an Academic Editor for PeerJ.

### Author Contributions

- Gabriel C. Coppo conceived and designed the experiments, performed the experiments, analyzed the data, prepared figures and/or tables, authored or reviewed drafts of the article, and approved the final draft.
- Araiene P. Pereira analyzed the data, prepared figures and/or tables, authored or reviewed drafts of the article, and approved the final draft.
- Sergio A. Netto performed the experiments, authored or reviewed drafts of the article, and approved the final draft.
- Angelo F. Bernardino conceived and designed the experiments, performed the experiments, authored or reviewed drafts of the article, and approved the final draft.

### Field Study Permissions

The following information was supplied relating to field study approvals (*i.e.*, approving body and any reference numbers):

Field sampling was authorized by the Biodiversity Authorization and Information System of the Brazilian Institute for the Environment and Renewable Natural Resources (sampling license: SISBIO-IBAMA 24700-1).

### Data Availability

The raw sequences data are available in NCBI SRA: SRR24675047.

## Supplemental Information

Supplemental information for this article can be found online at http://dx.doi.org/10.7717/peerj.17727#supplemental-information.

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
