# Peer review of "Meiofauna at a tropical sandy beach in the SW Atlantic: the influence of seasonality on diversity"

_PeerJ, doi:10.7717/peerj.17727_

## Round 0.1 · original submission · Major Revisions

Dear Dr. Coppo,

The three reviewers suggested that your manuscript needs a major review. Please review each comment carefully. I believe that your manuscript will be an important contribution to the area.

All the best

Juan Pablo Quimbayo

·

Basic reporting

1- English needs some improvement (see suggestions along the manuscript)
2- Literature references/citation need to be fixed.
3- The overall structure is good, but the authors should consider combining some of the tables or perhaps put them into the supplementary materials. Abiotic data could be presented as figures or supplementary figures (see comments).
4- Overall results align with hypothesis, but the ideas, especially in the discussion section, need to be better connected.
5- Lack of consistent: acronyms usage, writing style, abbreviations, etc.

Experimental design

1- The research fits with the aims and scope of the journal.
2- The research question/hypothesis is clear, and the findings contribute to the broader understanding of the benthic communities.
3- The methods section needs drastic improvement, particularly in the bioinformatics subsection. It is not clear how the replicates were combined, how the final volume used for DNA extraction was achieved, etc.
4- Some of the analysis need to better described in this section.
5- The spatial variation, i.e., variation among stations, was never treated here. What about patchiness?
6- Taxonomic assignment: the authors should consider to assign taxonomic using other methods (e.g., Blast+) to see if taxonomic resolution is improved.

Validity of the findings

Perhaps the most critical thing in this study is the variation in the number of reads obtained across samples, which currently it is not clear as the authors do not provide any summary from their bioinformatic analysis (e.g. Dada2 stats). This variation in the dataset should be dissected so that authors can have more evidence to support their ecological/biological conclusions. See comments along the manuscript.

Additional comments

The reviewer has made many suggestions/changes along the manuscript in the hope the authors can improve their work. The authors should address these concerns prior to resubmit the manuscript.

Reviewer 2 ·

Basic reporting

The data is unique and robust. The language is easy to follow.
There are some issues that deserve further attention.

Overall I think that introduction and discussion are quite long, this might be because the real scientific contribution of the article is underexplored.
So far, they just proved that meiofauna varied seasonally. They try to associatethe seasonal changes to the enviornmental changes, but the analytical apporach used is somehow limited.
Also their analytical approach and the hypothesis does not justify their title and the emphasis on the term seascapes dynamics.

I think that some rethinking on how the problematic is presentes and how the analysis was conducted are needed.

For instance, the intro needs some restructuring.
the problematic just appears at line 56, third paragraph, which seems to be the lack of knowledge in seasonal variation of sandy beach meiofauna.
Based on their text the only knowledge available on seasonality of meiofauna are the two studies coull 1988 and Albuquerque et al. 2007, which is not true. A quick search on the topic returned dozens of studies.
In my opinion this problematic has nothing to do with the title!

the paper does not make clear how they have integrated the concept of pelagic seascapes into their analysis.
Are the 20 stastions at the same seascape during a sampling period?
how did you handle the spatialvariability of the data retrived from the satellite?
how did you handle the spatialvariability of the benthic data?

Experimental design

The sampling methods and objectives are not coherent to me.
Based on their sampling I was expecting a modelling approach, using the environmental variables to predict/explain the biotic ones;
but their data analysis is a mixture of things. First they have described a test on a categorical variable "seasons" (which has been stated in the objectives). But for me, this is like tanking the whole potential of the dataset and ignoring it!!!!
If the seasons was the goal, why collecting every month? and all these env variables? and what about the spatial variation? the effort of processing 20 stations, each with 3 replicates and the 12 months were reduced to a permanova test of four seasons!!!!!
it is also not clear along the methods how the seascape variables or the seascapes itself (categories) were considered in the analysis (such as cap and linear model)

In addition, their modelling approach is quite simplistic

line 236: which type of linear model? the assumptions were met? were there highly correlated variables among the predictors?

there is a multitude o modelling approaches, why choosing the one that has several assumptions and limitations.

Validity of the findings

a R2 of 0.42 is quite low, don´t you think? There is still 68% of unexplained variance. I believe that such low r2 is a consequence of the chosen modelling approach.
How can you explain so much variance unexplained?
Based on this result, i concluded that the enviornmental variables that you have included in the model and the seascapes were not the major driving force structuring the meiofauna. Can you confirm that?

Additional comments

minor additional comments
abstract:
please provide further details about the study
the term seascape dataset and seascape dynamics are quite vague, provide the actual metrics

line 99: the test is on phylogenetic diversity and not on the standard measures, the reasons why were not clear along the introduction? and in fact much more is done along the results
line 99: how the seasonality of the seascape was measured?
line 100: what do you mean by larger-scale influence...this is quite vague for a test..
line 102: how this hypothesis is related to seascapes?
line 114: at which water depth the samples were taken?
line 122: what was the size of the sample size?
line 229: what is seascape coverage?

Reviewer 3 ·

Basic reporting

Coppo et al. used metabarcoding from sediment samples to assess the meiofaunal assemblage composition and diversity on a sandy beach in Southeast Brazil during one year (four samplings). The manuscript is interesting and generally well-written (but the connection between paragraphs can be improved to enhance the flow of information). I believe it deserves to be published; however, I have two main concerns that need to be addressed before publication:

1. The authors conducted an extremely poor revision of the literature. They aim to discuss sandy beach ecology, but they barely mention any sandy beach papers. They ignore virtually all the classic papers (e.g., Omar Defeo, Anthon McLachlan, Gerhard Masselink) and include only general references on different environments such as reefs, estuaries, and rhodolith beds (most of those references published by their group). Furthermore, out of almost 100 references, they only mention four works conducted in Brazil (except for their own work). This is surprising given that Brazil is one the most productive countries regarding sandy beach ecology, as shown in recent revisions published by Lercari (2023) and Corte et al. (2023). There are plenty of investigations that assessed seasonal variations on Brazilian sandy shores, including several related to climatic events/seascape dynamics. Most of them investigated macrofaunal assemblages, but meiofaunal studies are also available (e.g., Di Domenico et al. 2009)

Lercari D. (2023). Sandy beaches: publication features, thematic areas and collaborative networks between 2009 and 2019. Estuar Coast. Shelf Sci. 281, 108211. doi: 10.1016/j.ecss.2023.108211
Di Domenico, M., Da Cunha Lana, P. and Garraffoni, A.R.S. (2009), Distribution patterns of interstitial polychaetes in sandy beaches of southern Brazil. Marine Ecology, 30: 47-62. https://doi.org/10.1111/j.1439-0485.2008.00255.x
Corte et al. (2023) The science we need for the beaches we want: frontiers of the flourishing Brazilian ecological sandy beach research. https://doi.org/10.3389/fmars.2023.1200979

1. My second major concern relates to the main result of the paper. The authors found that meiofauna was dominated by Crustacea (46% of sequence reads) and Annelida (28% of sequence reads); however, these groups are typical of the macrobenthos. The most common meiofaunal group, Nematoda, accounted for only 12% of sequence reads. In the methods, they mention: “Here we considered meiofaunal metazoans all the exclusively meiofaunal phyla (Gnathostomulida, Kinorhyncha, Loricifera, Gastrotricha, and Tardigrada) and other metazoans that can be meiofaunal size during life (temporary meiofaunal taxa) (Higgins and Tiel, 1988; Giere, 2009), as previously realized in other studies (Brannock and Halanych, 2015; Bernardino et al., 2019; Coppo et al., 2023)”. Okay, other studies (mainly from their group – Bernardino et al. 2019 and Coppo et al. 2023) have done that, but this does not ensure the method is correct. Including organisms that may be considered meiofauna during their early stages does not imply they are meiofauna. Following their assumption, even adult macrobenthos would be considered meiofauna simply because they might have been considered meiofauna during a very short period of their life cycle. Moreover, I believe these species would release much more DNA (and be more detected) due to their larger body size, a fact that compromises the reliability of their results (please correct me if I am wrong). Therefore, I suggest the authors review their analysis by considering only exclusively meiofaunal phyla or using the more general term “benthic biodiversity” in their work. In the second scenario, they should review the literature to include both meio and macrofauna.

Specific comments:
L 101: Only one hypothesis is mentioned.

Methods
At what depth samples were collected?

Please indicate the dates samplings were collected.
Multiple comparisons are presented in Tables 2-6, but there is no mention of multiple comparison corrections.
Results
Tables – explain the acronyms

Discussion

The beginning of the discussion mainly repeats the results. Nothing is discussed (L347-371)
L372-373: “We observed that the dominant seascapes are associated to abundance of reads, Shannon’s diversity, and phylogenetic diversity patterns at local scale, supporting our initial hypothesis.”

Environmental data was more variable than biological assemblages, and the main predictors of benthic assemblages were biological variables. Thus, I am not 100% confident the results support the initial hypothesis.

Crustacea and Annelida are typically considered macrofaunal groups. How does size influence readings? I suppose larger individuals would be more easily detected.

L412-417: “Shannon’s Diversity index and phylogenetic diversity were positively influenced by the biopolymeric carbon (BPC) content, which represents the labile fraction of organic matter in sediment (Danovaro et al., 1993; Fabiano et al., 1995), showing that meiofaunal diversity is associated to food availability. Similarly, Cisneros et al. (2011) found observed seasonal changes on organic matter content and nutrients associated to differences on benthic abundance and diversity at a tropical sandy beach”

This is an interesting result that could be further explored. Sandy Beach biodiversity is known to be controlled by physical variables such as sediment and waves, but recent publications have shown that biological variables (e.g., food availability) may also be an important driver of beach macrobenthos (e.g., Rodil et al. 2012; Turra et al. 2014; Corte et al. 2022). Maybe this holds true for meio and macrofauna?

Rodil, I.F., Compton, T.J., Lastra, M., 2012. Exploring macroinvertebrate species distributions at regional and local scales across a sandy beach geographic continuum. PLoS ONE 7, e39609. https://doi.org/10.1371/journal.pone.0039609

Turra, A., Petracco, M., Amaral, A.C.Z., Denadai, M.R., 2014. Temporal variation in lifehistory traits of the clam Tivela mactroides (Bivalvia: Veneridae): density-dependent processes in sandy beaches. Estuar. Coast. Shelf Sci. 150, 157–164. https://doi.org/10.1016/j.ecss.2013.06.004.

Corte, G.N.; Checon, H.H.; Shah Esmaeili, Y.; Defeo, O.; Turra, A. 2022. Evaluation of the effects of urbanization and environmental features on sandy beach macrobenthos highlights the importance of submerged zones. Mar. Pol. Bul. 182: 113962

L423-431 – Nice job on highlighting the shortcomings. Can you also offer alternatives?

Conclusion

L-451-452: I would not say that one year is not a long-term scale


** Staff Note: Regarding suggested references, it is PeerJ policy that additional references suggested during the peer-review process should only be included if the authors are in agreement that they are relevant and useful **

Experimental design

No comment

Validity of the findings

See number 1

Additional comments

See number 1

---

## Round 0.2 · Major Revisions

Dear Dr. Coppo and co-authors

Thank you for waiting during this period. All three reviewers agree that the current version of your manuscript has shown significant improvement. However, they all suggest that your manuscript still requires careful revision. Therefore, I invite you to review the reviewers' comments and submit a new version of your manuscript. I believe that your work will be an important contribution to science.

All the best

Juan Pablo Quimbayo

·

Basic reporting

Overall, the authors improved the manuscript compared to the original version. The manuscript is now more concise and the findings more clear. On the other hand, the authors still need to consider:
1- English still needs improvement, 2- Be consistent with acronym and writing style throughout the manuscript, 3- Make sure that figures/tables represent the information given in the text. The reviewer made many comments along the manuscript which will hopefully help the authors to further improve their manuscript.

Also check for:
1- Make sure to better connect the paragraphs by always briefly introducing the next main idea
2- Make sure to be consistent with the verbal tense in the same sentence
3- Check for repetitive text/information
4- Why introduce acronyms if they are always written down in full
5- Figure 2 - order of variables according to the text in the results and make sure they match the text (high and/or low values in the right season)

Experimental design

The authors further explained how they processed their samples, from sampling to wet and lab protocols. Still, there were a few things left behind that the authors need to clarify (e.g., aliquots, total number of samples, why pooling things together, fragment size, etc.).

Validity of the findings

Overall, the findings (effects of seasonality on meiofauna diversity) are meaningful. Nevertheless, the authors concentrated their analyses only on meiofaunal metazoans and only phylum level. This approach (only part of dataset with coarse taxonomy) can certainly obscure some patterns (e.g., significant difference across ALL seasons). The authors should consider these in further analysis.

Additional comments

Table S1 shows that a lot of reads a been discard during the filtering step. A phred score of 39 is quite high. The authors should consider to play with the parameters used in Qiime/Dada2 to see if the recovery of reads in summer, autumn, and winter improve. Surprisingly, this very stringent approach did not affect the spring samples (higher number of reads).

One thing interesting is that, when only focusing on meiofaunal metazoans, spring samples displayed the lowest number of reads, so they went from the highest read counts to the lowest read counts. See comments on the resample step.

Reviewer 2 ·

Basic reporting

Overall, its is a high quality data set with relevant information. The reviewed version has improved, but some points are still not clear to me. The response letter could be better presented, since they copied and pasted the same answer to different comments.
In this new verion, I still have difficulties to understand the novelty of this study, lack of sandy beaches studies should not be used as a novelty. I am also not conviced that they have performed the best analytical approach.
Below, I give some more comments to the authros that may help them to improve the quality of the study.

Experimental design

DAta analysis
As already stated in the previous review, they put their beautiful monthly data set spatially replicated into the "bin" when reducing it to seasonal variation. In summary I do not agree with their choice of using permanova.
Note that monthly variation, seasonal variation and spatial variation can be analysed simuntaneously. Even more interesting, they had additionally collected the enviornmental data, meaning that they could analyse the whole data just against the environmental conditions and then project it along the seasons. The organisms do not percieve the calendar but may percieve changes in temperature.
In line 302 the proposed analysis is somehow more in the direction of my arguments, althoug linear models are not the best choices to explore the patterns of biodiveristy. The same rational could be done for the multivariate data. The cited referecences at this section explore the difference between permanova and regression methods.

Validity of the findings

no comments

Additional comments

abstract
The concept of seascape as temperature, salinity and other variables is not all correct. these are variables that could be used to describe a seascape but they are not the seascape per se!!!

methods: this is not what you have done (calculate the diversity over the year) please refer to your hypothesis testing at this section of the abstract

results: please correct "meiofauna diversity is not driven by ...."
What do you mean by complex set?
this sentence may be influenced by ecological interactions among taxa is neiter a result nor a conclusion of your study, so I suggest to remove it.


Introduction
Much too long. Please concentrate to the real problem and scientific advance. There is no need of 9 paragraphs to explore the topic. I would suggest to reduce it to 4/5 paragraphs.
lines 48-52: sentence too long, some edition is recommended!
line 93-106: this paragraph makes a mixtures of scales...it starts with latitudinal gradient, regional patterns and local responses. Why do you have to present all the scales here.
lines 100-106: show us how they have changed: increased or decreased????
lines 107-113: so what?
lines 114-133: this paragraph is completlely out of context
139: so it is not all original? here is a publication which is intended to present a scientific advance. this has to be put clearly along the introduction.
146-150: the first hypothesis was some how argumented at lines 100-106, but the second hypothesis was not supported along the introduction. If the authors want to make a parallel of meiofauna patterns with those of benthic recruitment, this has to be clearly stated and preferably supported by the literature that this parallel is reasonable.


Results
table 2
there is no need to present the pairwise comparisons of the interaction when no significance effect was detected

Discussion
Lines 398-403 this type of info should be placed in the introduction
lines 442: it can be that macrofauna dna is present in the samples, justifying the differences in relative abundance encoutered in this study?
lines 490-497: plabe along the introduction!

Reviewer 3 ·

Basic reporting

Overall, I like the manuscript and believe it deserves to be published. Sandy beach ecosystems are understudied, and the authors bring a new perspective to study beach biodiversity with eDNA. The inclusion of sandy beach references improved both the Introduction and Discussion. However, in my opinion, the authors are pushing too hard attempting to discuss seascapes but sampling one sandy beach. I suggest focusing on the paper's main results: seasonal changes in meiofaunal communities on one sandy beach. It has great merit, especially since this is the first metabarcoding assessment of benthic animals in the area

Introduction
The introduction is too long and goes back and forth (sandy beach – environment – meiofauna – environment – meiofauna – environment).
L51: Remove "and” before "is crucial"
L134: "Predicting changes in diversity patterns from local to global scales is a paramount in a scenario of global environmental change" I agree, but...
L139: Portions of this text were previously published as part of a thesis (Coppo, 2023)????
Methods
L210: Seascape characterization?
With all due respect, this language feels like buying a transatlantic to play in a bathtub.
Results
I am still worried about the dominance of polychaetes and crustaceans, but I accept the author's explanation.
Discussion
The Discussion and Introduction are much improved by including sandy beach references.

Experimental design

No comment

Validity of the findings

No comment

---

## Round 0.3 · Minor Revisions

Dear Dr. Coppo

Please review the last comments before your manuscript is accepted in the journal.

All the best,

Juan Pablo

·

Basic reporting

The authors have addressed all comments/suggestions, thus improving significantly the manuscript.

Experimental design

The authors have clarified all aspects related to sampling and sample processing in this revised version.

Validity of the findings

The data and analysis support the findings/conclusions made by the authors. In this revised version, the discussion is much clear.

Reviewer 3 ·

Basic reporting

I appreciate the reorganization of the manuscript. The introduction is shorter and easier to follow, and I congratulate the authors for their nice work. Overall, I believe this manuscript deserves to be published after a few minor modifications.

Abstract
L15- The diversity, distribution and composition of what? Consider using “Their diversity”

Introduction
L66 – I suggest replacing “Nonetheless, seasonal variation has been less investigated” with “ Accordingly/Consequently, seasonal variation has been overlooked/understudied”

L 95 – “ Meiofaunal taxa may have specific adaptations and respond differently to environmental conditions”
Differently from what?

L 107- Which patterns? Spatial? Temporal?

L108 – Replace “ understanding how it changes over time and predicting how it is going to happen in the future” with “ investigating variations over time and predicting future changes”

L116- add “ and” before the second goal (ii)

Discussion

L 362 - add “and” before “understanding”

L368 – Corte et al. 2022 sampled almost one hundred beach sites in SE Brazil and found that “Microphytobenthos content, a proxy for food availability, was also an important correlate of macrobenthic assemblages' metrics. (…) Our findings therefore reinforce increasing evidence of the relevance of food supply in beach ecosystems”.
Considering that this study is one of the most comprehensive field evaluations of the drivers of sandy beach biodiversity, one of the few studies highlighting the importance of biological factors/food availability for sandy beach benthic diversity, and was performed close to your study area, I strongly suggest including this reference.

Corte et al. 2022 - https://doi.org/10.1016/j.marpolbul.2022.113962

L429-430 – “These differences in diversity metrics shows that, at Gramuté beach, meiofaunal diversity changes seasonally but is not different among dry and rainy periods” Please explain what you mean by “it changes but is not different” (also these differences show).

Experimental design

no comment

Validity of the findings

no comment

Additional comments

no comment

---

## Round 0.4 · accepted · Accept

Dear Dr. Coppo,

Thanks for your work during the reviewer process. I would like to communicate that your manuscript was accepted.

All the best,

Juan Pablo